# Visual Concept Learning: Combining Machine Vision and Bayesian Generalization on Concept Hierarchies

**Yangqing Jia[1], Joshua Abbott[2], Joseph Austerweil[3], Thomas Griffiths[2], Trevor Darrell[1]**
[1]UC Berkeley EECS     [2]Dept of Psychology, UC Berkeley
[3]Dept of Cognitive, Linguistics, and Psychological Sciences, Brown University
{jiayq, joshua.abbott, tom_griffiths, trevor}@berkeley.edu
joseph_austerweil@brown.edu

## Abstract

Learning a visual concept from a small number of positive examples is a significant challenge for machine learning algorithms. Current methods typically fail to find the appropriate level of generalization in a concept hierarchy for a given set of visual examples. Recent work in cognitive science on Bayesian models of generalization addresses this challenge, but prior results assumed that objects were perfectly recognized. We present an algorithm for learning visual concepts directly from images, using probabilistic predictions generated by visual classifiers as the input to a Bayesian generalization model. As no existing challenge data tests this paradigm, we collect and make available a new, large-scale dataset for visual concept learning using the ImageNet hierarchy as the source of possible concepts, with human annotators to provide ground truth labels as to whether a new image is an instance of each concept using a paradigm similar to that used in experiments studying word learning in children. We compare the performance of our system to several baseline algorithms, and show a significant advantage results from combining visual classifiers with the ability to identify an appropriate level of abstraction using Bayesian generalization.

## 1   Introduction

Machine vision methods have achieved considerable success in recent years, as evidenced by performance on major challenge problems [4, 7], where strong performance has been obtained for assigning one of a large number of labels to each of a large number of images. However, this research has largely focused on a fairly narrow task: assigning a label (or sometimes multiple labels) to a single image at a time. This task is quite different from that faced by a human child trying to learn a new word, where the child is provided with multiple positive examples and has to generalize appropriately. Even young children are able to learn novel visual concepts from very few positive examples [3], something that still poses a challenge for machine vision systems. In this paper, we define a new challenge task for computer vision – *visual concept learning* – and provide a first account of a system that can learn visual concepts from a small number of positive examples.

In our visual concept learning task, a few example images from a visual concept are given and the system has to indicate whether a new image is or is not an instance of the target concept. A key aspect of this task is determining the degree to which the concept should be generalized [21] when multiple concepts are logically consistent with the given examples. For example, consider the concepts represented by examples in Figure 1 (a-c) respectively, and the task of predicting whether new images (d-e) belong to them or not. The ground truth from human annotators reveals that the level of generalization varies according to the conceptual diversity, with greater diversity leading to broader generalization. In the examples shown in Figure 1, people might identify the concepts as (a) Dalmatians, (b) all dogs, and (c) all animals, but not generalize beyond these levels although no

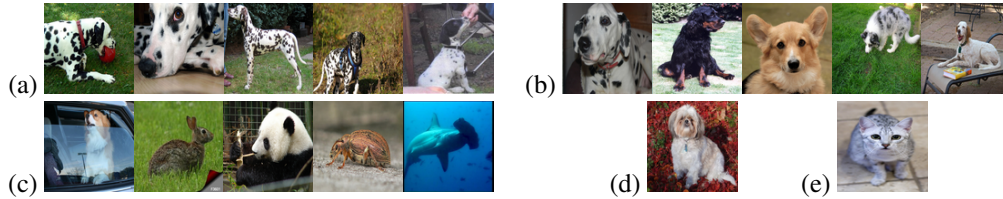

Figure 1: Visual concept learning. (a-c): positive examples of three visual concepts. Even without negative data, people are able to learn these concepts: (a) Dalmatians, (b) dogs and (c) animals. Note that although (a) contains valid examples of dogs and both (a) and (b) contain valid examples of animals, people restrict the scope of generalization to more specific concepts, and find it easy to make judgments about whether novel images such as (d) and (e) are instances of the same concepts – the task we refer to as *visual concept learning*.

negative images forbids so. Despite recent successes in large-scale category-level object recognition, we will show state-of-the-art machine vision systems fail to exhibit such patterns of generalization, and have great difficulty learning without negative examples.

Bayesian models of generalization [1, 18, 21] account for these phenomena, determining the scope of a novel concept (e.g., does the concept refer to Dalmatians, all dogs, or all animals?) in a similar manner to people. However, these models were developed by cognitive scientists interested in analyzing human cognition, and require examples to be manually labeled as belonging to a particular leaf node in a conceptual hierarchy. This is reasonable if one is asking whether proposed psychological models explain human behavior, but prevents the models from being used to automatically solve visual concept learning problems for a robot or intelligent agent.

We bring these two threads of research together, using machine vision systems to assign novel images locations within a conceptual hierarchy and a Bayesian generalization model to determine how to generalize from these examples. This results in a system that comes closer to human performance than state-of-the-art machine vision baselines. As an additional contribution, since no existing dataset adequately tests human-like visual concept learning, we have collected and made available to the community the first large-scale dataset for evaluating whether machine vision algorithms can learn concepts that agree with human perception and label new unseen images, with ground-truth labeling obtained from human annotators from Amazon Mechanical Turk. We believe that this new task provides challenges beyond the conventional object classification paradigms.

## 2 Background

In machine vision, scant attention has been given to the problem of learning a visual concept from a few positive examples as we have defined it. When the problem has been addressed, it has largely been considered from a hierarchical regularization [16] or transfer learning [14] perspective, assuming that a fixed set of labels are given and exploiting transfer or regularization within a hierarchy. Mid-level representations based on attributes [8, 13] focus on extracting common attributes such as "fluffy" and "aquatic" that could be used to semantically describe object categories better than low-level features. Transfer learning approaches have been proposed to jointly learn classifiers with structured regularization [14].

Of all these previous efforts, our paper is most closely related to work that uses object hierarchies to support classification. Salakhutdinov et al. [16] proposed learning a set of object classifiers with regularization using hierarchical knowledge, which improves the classification of objects at the leaves of the hierarchy. However, this work did not address the problem of determining the level of abstraction within the hierarchy at which to make generalizations, which is a key aspect of the visual concept learning problem. Deng et al. [5] proposed predicting object labels only to a granularity that the classifier is confident with, but their goal was minimizing structured loss rather than mimicking human generalization.

Existing models from cognitive science mainly focus on understanding human generalization judgments within fairly restricted domains. Tenenbaum and colleagues [18, 20] proposed mathematical abstractions for the concept learning problem, building on previous work on models of generalization by Shepard [17]. Xu and Tenenbaum [21] and Abbott et al. [1] conducted experiments

with human participants that provided support for this Bayesian generalization framework. Xu and Tenenbaum [21] showed participants one or more positive examples of a novel word (e.g., "these three objects are Feps"), while manipulating the taxonomic relationship between the examples. For instance, participants could see three toy Dalmatians, three toy dogs, or three toy animals. Participants were then asked to identify the other "Feps" among a variety of both taxonomically related and unrelated objects presented as queries. If the positive examples were three Dalmatians, people might be asked whether other Dalmatians, dogs, and animals are Feps, along with other objects such as vegetables and vehicles. Subsequent work has used the same basic methodology in experiments using a manually collated set of images as stimuli [1].

All of these models assume that objects are already mapped onto locations in a perceptual space or conceptual hierarchy. Thus, they are not able to make predictions about genuinely novel stimuli. Linking such generalization models to direct perceptual input is necessary in order to be able to use this approach to learn visual concepts directly from images.

## 3 A Large-scale Concept Learning Dataset

Existing datasets (PASCAL [7], ILSVRC [2], etc.) test supervised learning performance with relatively large amounts of positive and negative examples available, with ground truth as a set of mutually-exclusive labels. To our knowledge, no existing dataset accurately captures the task we refer to as visual concept learning: to learn a novel word from a small set of positive examples like humans do. In this section, we describe in detail our effort to make available a dataset for such task.

### 3.1 Test Procedure

In our test procedure, an agent is shown $n$ example images ($n = 5$ in our dataset) sampled from a node (may be leaf nodes or intermediate nodes) from the ImageNet synset tree, and is then asked whether other new images sampled from ImageNet belong to the concept or not. The scores that the agent gives are then compared against human ground truth that we collect, and we use precision-recall curves to evaluate the performance.

From a machine vision perspective, one may ask whether this visual concept learning task differs from the conventional ImageNet-defined classification problem – identifying the node from which the examples are drawn, and then answering yes for images in the subtree corresponding to the node, and no for images not from the node. In fact, we will show in Section 5.2 that using this approach fails to explain how people learn visual concepts. Human performance in the above task exhibits much more sophisticated concept learning behaviors than simply identifying the node itself, and the latter differs significantly from what we observe from human participants. In addition, with no negative images, a conventional classification model fails to distinguish between nodes that are both valid candidates (e.g., "dogs" and "animals" when shown a bunch of dog images). These make our visual concept learning essentially different and richer than a conventional classification problem.

### 3.2 Automatic Generation of Examples and Queries

Large-scale experimentation requires an efficient scheme to generate test data across varying levels of a concept hierarchy. To this end, we developed a fully-automated procedure for constructing a large-scale dataset suitable for a challenge problem focused on visual concept learning. We used the ImageNet LSVRC [2] 2010 data as the basis for automatically constructing a hierarchically-organized set of concepts at four different levels of abstraction. We had two goals in constructing the dataset: to cover concepts at various levels of abstraction (from subordinate concepts to super-ordinate concepts, such as from Dalmatian to living things), and to find query images that comprehensively test human generalization behavior. We address these two goals in turn.

To generate concepts at various levels of abstraction, we use all the nodes in the ImageNet hierarchy as concept candidates, starting from the leaf node classes as the most specific level concept. We then generate three more levels of increasingly broad concepts along the path from the leaf to the root for each leaf node in the hierarchy. Examples from such concepts are then shown to human participants to obtain human generalization judgements, which will serve as the ground truth. Specifically, we use the leaf node class itself as the most basic trial type $L_0$, and select three levels of nested concepts

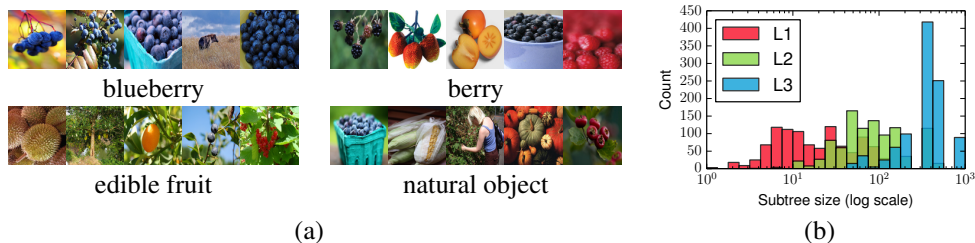

blueberry      berry

edible fruit      natural object

(a)          (b)

Figure 2: Concepts drawn from ImageNet. (a) example images sampled from the four levels for `blueberry`, and (b) the histogram for the subtree sizes of different levels of concepts (x axis in log scale).

$L_1$, $L_2$, $L_3$ which correspond to three intermediate nodes along the path from the leaf node to the root. We choose the three nodes that maximize the combined information gain across these levels:

$$\mathcal{C}(L_{1\cdots3}) = \sum\nolimits_{i=0}^{3} \log(|L_{i+1}| - |L_i|) - \log|L_{i+1}|, \tag{1}$$

where $|L_i|$ is the number of leaf nodes under the subtree rooted at $L_i$, and $L_4$ is the whole taxonomy tree. As a result, we obtain levels that are "evenly" distributed over the taxonomy tree. Such levels coarsely correspond to the sub-category, basic, super-basic, and super-category levels in the taxonomy: for example, the four levels used in Figure 1 are `dalmatian`, `domestic dog`, `animal`, `organism` for the leaf node `dalmatian`, and in Figure 2(a) are `blueberry`, `berry`, `edible fruit`, and `natural object` for the leaf node `blueberry`. Figure 2(b) shows a histogram of the subtree sizes for $L_1$ to $L_3$ respectively.

For each concept, the five images shown to participants as examples of that concept were randomly sampled from five different leaf node categories from the corresponding subtree in the ILSVRC 2010 test images. Figure 1 and 2 show such examples.

To obtain the ground truth (the concepts people perceive when given the set of examples), we then randomly sample twenty query images, and ask human participants whether each of these query images belong to the concept given by the example images. A total of 20 images are randomly sampled as follows: three each from the $L_0$, $L_1$, $L_2$ and $L_3$ subtrees, and eight images outside $L_3$. This ensures a complete coverage over in-concept and out-of-concept queries. We explicitly made sure that the leaf node classes of the query images were different from those of the examples if possible, and no duplicates exist among the 20 queries. Note that we always sampled the example and query images from the ILSVRC 2010 test images, allowing us to subsequently train our machine vision models with the training and validation images from the ILSVRC dataset while keeping those in the visual concept learning dataset as novel test images.

### 3.3 Collecting Human Judgements

We created 4,000 identical concepts (four for each leaf node) using the protocol above, and recruited participants online through Amazon Mechanical Turk (AMT, `http://www.mturk.com`) to obtain the human ground truth data. For each concept, an AMT HIT (a single task presented to the human participants) is formed with five example images and twenty query images, and the participants were asked whether each query belongs to the concept represented by the examples. Each HIT was completed by five unique participants, with a compensation of $0.05 USD per HIT. Participants were allowed to complete as many unique trials as they wished. Thus, a total of 20,000 AMT HITs were collected, and a total of 100,000 images were shown to the participants. On average, each participant took approximately one minute to finish each HIT, spending about 3 seconds per query image. The dataset is publicly available at `http://www.eecs.berkeley.edu/~jiayq/`.

## 4 Visually-Grounded Bayesian Concept Learning

In this section, we describe an end-to-end framework which combines Bayesian word learning models and visual classifiers, and is able to perform concept learning with perceptual inputs.

## 4.1 Bayesian Concept Learning

Prior work on concept learning [21] addressed the problem of generalization from examples using a Bayesian framework: given a set of $N$ examples (images in our case) $\mathcal{X} = \{\mathbf{x}_1, \mathbf{x}_2, \ldots, \mathbf{x}_N\}$ that are members of an unknown concept $\mathcal{C}$, the probability that a query instance $\mathbf{x}_{\text{query}}$ also belongs to the same concept is given by

$$P_{\text{new}}(\mathbf{x}_{\text{query}} \in \mathcal{C}|\mathcal{X}) = \sum_{h \in \mathcal{H}} P_{\text{new}}(\mathbf{x}_{\text{new}}|h)P(h|\mathcal{X}), \tag{2}$$

where $\mathcal{H}$ is called the "hypothesis space" – a set of possible hypotheses for what the concept might be. Each hypothesis corresponds to a (often semantically related) subset of all the objects in the world, such as "dogs" or "animals". Given a specific hypothesis $h$, the probability $P_{\text{new}}(x_{\text{new}}|h)$ that a new instance belongs to it is 1 if $x_{\text{new}}$ is in the set, and 0 otherwise, and $P(h|\mathcal{X})$ is the *posterior* probability of a hypothesis $h$ given the examples $\mathcal{X}$.

The posterior distribution over hypotheses is computed using the Bayes' rule: it is proportional to the product of the *likelihood*, $P(\mathcal{X}|h)$, which is the probability of drawing these examples from the hypothesis $h$ uniformly at random times the *prior* probability $P(h)$ of the hypothesis:

$$P(h|\mathcal{X}) \propto P(h) \prod_{i=1}^{N} P_{\text{example}}(\mathbf{x}_i|h), \tag{3}$$

where we also make the strong sampling assumption that each $\mathbf{x}_i$ is drawn uniformly at random from the set of instances picked out by $h$. Importantly, this ensures that the model acts in accordance with the "size principle" [18, 20], meaning that the conditional probability of an instance given a hypothesis is inversely proportional to the size of the hypothesis, i.e., the number of possible instances that could be drawn from the hypothesis:

$$P_{\text{example}}(\mathbf{x}_i|h) = |h|^{-1} I(\mathbf{x}_i \in h), \tag{4}$$

where $|h|$ is the size of the hypothesis and $I(\cdot)$ is an indicator function that has value 1 when the statement is true. We note that the probability of an *example* and that of a *query* given a hypothesis are different: the former depends on the size of the underlying hypothesis, representing the nature of training with strong sampling. For example, as the number of examples that are all Dalmatians increases, it becomes increasingly likely that the concept is just Dalmatians and not dogs in general even though both are logically possible, because it would have been incredibly unlikely to only sample Dalmatians given that the truth concept was dogs. In addition, the prior distribution $P(h)$ captures biases due to prior knowledge, which favor particular kinds of hypotheses over others (which we will discuss in the next subsection). For example, it is known that people favor basic level object categories such as dogs over subcategories (such as Dalmatians) or supercategories (such as animals).

## 4.2 Concept Learning with Perceptual Uncertainty

Existing Bayesian word learning models assume that objects are perfectly recognized, thus representing them as discrete indices into a set of finite tokens. Hypotheses are then subsets of the complete set of tokens and are often hierarchically nested. Although perceptual spaces were adopted in [18], only very simple hypotheses (rectangles over the position of dots) were used. Performing Bayesian inference with a complex perceptual input such as images is thus still a challenge. To this end, we utilize the state-of-the-art image classifiers and classify each image into the set of leaf node classes given in the ImageNet hierarchy, and then build a hypothesis space on top of the classifier outputs.

Specifically, we construct the hypothesis space over the image labels using the ImageNet hierarchy, with each subtree rooted at a node serving as a possible hypothesis. The hypothesis sizes are then computed as the number of leaf node classes under the corresponding node, e.g., the node "animal" would have a larger size than the node "dogs". The large number of images collected by ImageNet allows us to train classifiers from images to the leaf node labels, which we will describe shortly. Assuming that there are a total of $K$ leaf nodes, for an image $\mathbf{x}_i$ that is classified as label $\hat{y}_i$, the likelihood $P(\mathbf{x}_i|h)$ is then defined as

$$P_{\text{example}}(\mathbf{x}_i|h) = \sum_{j=1}^{K} A_{j\hat{y}_i} \frac{1}{|h|} I(j \in h), \tag{5}$$

where $\mathbf{A}$ is the normalized confusion matrix, with $A_{j,i}$ being the probability that the true leaf node is $j$ given the classifier output being $i$. The motivation of using the confusion matrix is that classifiers are not perfect and misclassification could happen. Thus, the use of the confusion matrix incorporates the visual ambiguity into the word learning framework by providing an unbiased estimation of the true leaf node label for an image.

The prior probability of a hypothesis was defined to be an Erlang distribution, $P(h) \propto (|h|/\sigma^2) \exp\{-|h|/\sigma\}$, which is a standard prior over sizes in Bayesian models of generalization [17, 19]. The parameter $\sigma$ is set to 200 according to [1] in order to fit human cognition, which favors basic level hypotheses [15]. Finally, the probability of a new instance belonging to a hypothesis is similar to the likelihood, but without the size term, as $P_{\text{new}}(\mathbf{x}_{\text{new}}|h) = \sum_{j=1}^{K} A_{j\hat{y}_{\text{new}}} I(\hat{y}_{\text{new}} \in h)$, where $\hat{y}_{\text{new}}$ is the classifier prediction.

### 4.3   Learning the Perceptual Classifiers

To train the image classifiers for the perceptual component in our model, we used the ILSVRC training images, which consisted of 1.2 million images categorized into the 1,000 leaf node classes, and followed the pipeline in [11] to obtain feature vectors to represent the images. This pipeline uses 160K dimensional features, yielding a total of about 1.5TB for the training data. We trained the classifiers with linear multinomial logistic regressors with minibatch Adagrad [6] algorithm, which is a quasi-Newton stochastic gradient descent approach. The hyperparameters of the classifiers are learned with the held-out validation data.

Overall, we obtained a performance of 41.33% top-1 accuracy and a 61.91% top-5 accuracy on the validation data, and 41.28% and 61.69% respectively on the testing data, and the training took about 24 hours with 10 commodity computers. Although this is not the best ImageNet classifier to date, we believe that the above pipeline is a fair representation of the state-of-the-art computer vision approaches. Algorithms using similar approaches have reported competitive performance in image classification on a large number of classes (on the scale of tens of thousands) [10, 9], which provides reassurance about the possibility of using state-of-the-art classification models in visual concept learning.

To obtain the confusion matrix $\mathbf{A}$ of the classifiers, we note that the validation data alone does not suffice to provide a dense estimation of the full confusion matrix, because there is a large number of entries (1 million) but very few validation images (50K). Thus, instead of using the validation data for estimation of $\mathbf{A}$, we approximated the classifier's leave-one-out (LOO) behavior on the training data with a simple one-step gradient descent update to "unlearn" each image. Specifically, we started from the trained classifier parameters, and for each training image $\mathbf{x}$, we compute the gradient of the loss function when $\mathbf{x}$ is left out of the training set. We then take one step update in the direction of the gradient to obtain the updated classifier, and use it to perform prediction on $\mathbf{x}$. This allows us to obtain a much denser estimation that worked better than existing methods. We refer the reader to the supplementary material for the technical details about the classifier training and the LOO confusion matrix estimation.

## 5   Experiments

In this section, we describe the experimental protocol adopted to compare our system with human performance and compare our system against various baseline algorithms. Quantitatively, we use the precision-recall curve, the average precision (AP) and the $F_1$ score at the point where precision = recall to evaluate the performance and to compare against the human performance, which is calculated by randomly sampling one human participant per distinctive HIT, and comparing his/her prediction against the four others.

To the best of our knowledge, there are no existing vision models that explicitly handles our concept learning task. Thus, we compare our vision baseg Bayes generalization algorithm (denoted by **VG**) described in the previous section against the following baselines, which are reasonable extensions of existing vision or cognitive science models:

1. **Naive vision approach** (NV): this uses a nearest neighbor approach by computing the score of a query as its distance to the closest example image, using GIST features [12].

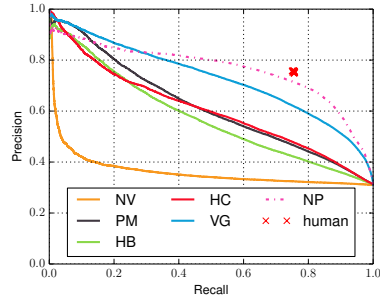

| Method | AP | $F_1$ Score |
|---|---|---|
| NV | 36.37 | 35.64 |
| PM | 61.74 | 56.07 |
| HC | 60.58 | 56.82 |
| HB | 57.50 | 52.72 |
| NP | 76.24 | 72.70 |
| **VG (ours)** | 72.82 | 66.97 |
| Human Performance | - | 75.47 |

Figure 3: The precision-recall curves of our method and the baseline algorithms. The human results are shown as the red crosses, and the non-perceptual Bayesian word learning model (NB) is shown as magenta dashed lines. The table summarizes the average precision (AP) and $F_1$ scores of the methods.

2. **Prototype model** (PM): an extension of the image classifiers. We use the $L_1$ normalized classifier output from the multinomial logistic regressors as a vector for the query image, and compute the score as its $\chi^2$ distance to the closest example image.

3. **Histogram of classifier outputs** (HC): similar to the prototype model, but instead of computing the distance between the query and each example, we compute the score as the $\chi^2$ distance to the histogram of classifier outputs, aggregated over the examples.

4. **Hedging the bets extension** (HB): we extend the hedging idea [5] to handle sets of query images. Specifically, we find the subtree in the hierarchy that maximizes the information gain while maintaining an overall accuracy above a threshold $\epsilon$ over the set of example images. The score of a query image is then computed as the probability that it belongs to this subtree. The threshold $\epsilon$ is tuned on a randomly selected subset of the data.

5. **Non-perceptual word learning** (NP): the classical Bayesian word learning model in [21] assuming a perfect classifier, i.e., by taking the ground-truth leaf labels for the test images. This is not practical in actual applications, but evaluating NP helps understand how a perceptual component contributes to modeling human behavior.

## 5.1 Main Results

Figure 3 shows the precision-recall curves for our method and the baseline methods, and summarizes the average precision and $F_1$ scores. Conventional vision approaches that build upon image classifiers work better than simple image features (such as GIST), which is sensible given that object categories provide relatively more semantics than simple features. However, all the baselines still have performances far from human's, because they miss the key mechanism for inferring the "width" of the latent concept represented by a set of images (instead of a single image as conventional approaches assume). In contrast, adopting the size principle and the Bayesian generalization framework allows us to perform much better, obtaining an increase of about 10% in average precision and $F_1$ scores, closer to the human performance than other visual baselines.

The non-perceptual (NP) model exhibits better overall average precision than our method, which suggests that image classifiers can still be improved. This is indeed the case, as state-of-the-art recognition algorithms may still significantly underperform human. However, note that for a system to work in a real-world scenario such as aid-giving robots, it is crucial that the agent be able to take direct perceptual inputs. It is also interesting to note that all visual models yield higher precision values in the low-recall region (top left of Figure 3) than the NP model, which does not use perceptual input and has a lower starting precision. This suggests that perceptual signals do play an important role in human generalization behaviors, and should not be left out of the pipeline as previous Bayesian word learning methods do.

## 5.2 Analysis of Per-level Responses

In addition to the quantitative precision-recall curves, we perform a qualitative per-level analysis similar to previous word learning work [1]. To this end, we binarize the predictions at the threshold that yields the same precision and recall, and then plot the per-level responses, i.e., the proportion of query images from level $L_i$ that are predicted positive, given examples from level $L_j$.

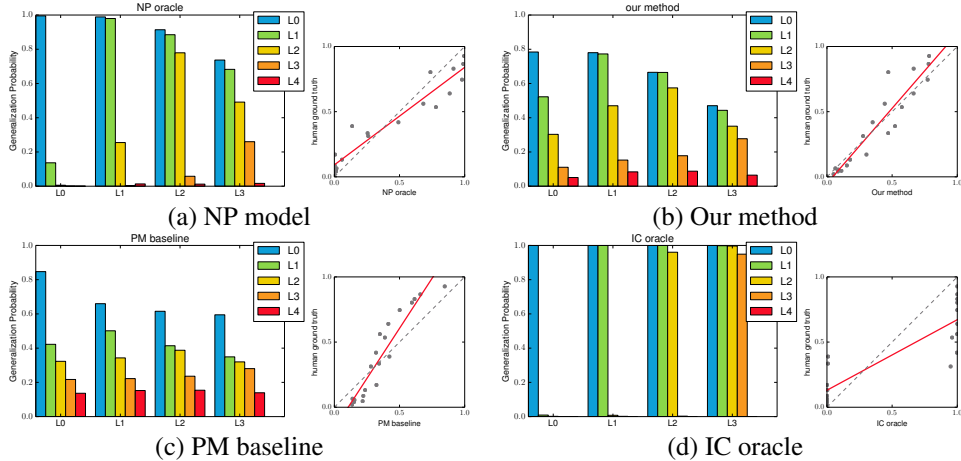

(a) NP model      (b) Our method

(c) PM baseline      (d) IC oracle

Figure 4: Per-level generalization predictions from various methods, where the horizontal axis shows four levels at which examples were provided ($L_0$ to $L_3$). At each level, five bars show the proportion of queries form levels $L_0$ to $L_4$ that are labeled as instances of the concept by each method. These results are summarized in a scatter plot showing model predictions (horizontal axis) vs. human judgments (vertical axis), with the red line showing a linear regression fit.

We show in Figures 4 and 5 the per-level generalization results from human, the NP model, our method, and the PM baseline which best represents state-of-the-art vision baselines. People show a monotonic decrease in generalization as the query level moves conceptually further from the examples. In addition, for queries of the same level, its generalization score peaks when examples from the same level are presented, and drops when lower or higher level examples are presented. The NP model tends to give more extreme predictions (either very low or very high), possibly due to the fact that it assumes perfect recognition, while visual inputs are actually difficult to precisely

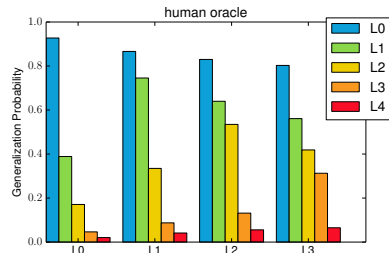

Figure 5: Per-level generalization from human participants.

classify even for a human being. The conventional vision baseline does not utilize the size principle to model human concept learning, and as a result shows very similar behavior with different level of examples. Our method exhibits a good correlation with the human results, although it has a smaller generalization probability for $L_0$ queries, possibly because current visual models are still not completely accurate in identifying leaf node classes [5].

Last but not least, we examine how well a conventional image classification approach could explain our experimental results. To do so, Figure 44(d) plots the results of an image classification (IC) oracle that predicts yes for an image within the ground-truth ImageNet node that the current examples were sampled from and no otherwise. Note that the IC oracle never generalizes beyond the level from which the examples are drawn, and thus, exhibits very different generalization results compared to the human participants in our experiment. Thus, visual concept learning poses more realistic and challenging problems for computer vision studies.

## 6 Conclusions

We proposed a new task for machine vision – visual concept learning – and presented the first system capable of approaching human performance on this problem. By linking research on object classification in machine vision and Bayesian generalization in cognitive science, we were able to define a system that could infer the appropriate scope of generalization for a novel concept directly from a set of images. This system outperforms baselines that draw on previous approaches in both machine vision and cognitive science, coming closer to human performance than any of these approaches. However, there is still significant room to improve performance on this task, and we present our visual concept learning dataset as the basis for a new challenge problem for machine vision, going beyond assigning labels to individual objects.

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
