[Supplementary Material]

# Visual Concept Learning: Combining Machine Vision and Bayesian Generalization on Concept Hierarchies Supplementary Materials

**Yangqing Jia**[1]**, Joshua Abbott**[2]**, Joseph Austerweil**[3]**, Thomas Griffiths**[2]**, Trevor Darrell**[1]

[1]UC Berkeley EECS    [2]Dept of Psychology, UC Berkeley
[3]Dept of Cognitive, Linguistics, and Psychological Sciences, Brown University
{jiayq, joshua.abbott, tom_griffiths, trevor}@berkeley.edu
joseph_austerweil@brown.edu

## Abstract

In the supplementary materials, we present more details to the data collection procedure, and more details about the training of our image classification component, including the large-scale classifier training and the confusion matrix estimation.

## 1  Data Collection

To construct the large-scale concept learning dataset, participants that were recruited online through Amazon Mechanical Turk were presented with a display that contained example images of a concept and query images, where participants could easily click what query images belong to the given category or not. Following previous work [1, 7], participants were told that "Mr. Frog" had picked out some examples of a word in a different language, and that "Mr. Frog" needed help picking out the other objects that could be called that word (see Figure 1 for the precise wording). Figure 1 shows an example display that a participant could have seen, and Figure 2 shows a possible response from a participant for a given trial. See the main text for further details about the experiment.

## 2  More on Training Large-scale Classifiers

We will elaborate a little more on how we learn the classifiers, that gives the prediction $\hat{y}$ of an image as

$$\hat{y} = f(\mathbf{x}) = \operatorname{argmax}_j \quad \boldsymbol{\theta}_j^\top \mathbf{x}, \tag{1}$$

where $1 < j < K$ where $K$ is the number of classes, and $\{\boldsymbol{\theta}\}_1^K$ are the classifier outputs. Basically, we focus on training large-scale linear multinomial logistic regressors, which optimizes the following objective function when given a set of training data:

$$\mathcal{L}(\boldsymbol{\theta}) = \lambda\|\boldsymbol{\theta}\|_2^2 - \sum_{i=1}^M \mathbf{t}_i \log \mathbf{u}_i, \tag{2}$$

where $\mathbf{t}_i$ is a 0-1 indicator vector where only the $y_i$-th element is 1, and $\mathbf{u}_i$ is the softmax of the linear outputs

$$u_{ij} = \exp(\boldsymbol{\theta}_j^\top \mathbf{x}_i)/\sum_{j'=1}^K \exp(\boldsymbol{\theta}_{j'}^\top \mathbf{x}_i), \tag{3}$$

where $\mathbf{x}_i$ is the feature for the $i$-th training image.

To perform training, for each iteration $t$ we randomly sample a minibatch from the data to estimate the gradient $\mathbf{g}_t$, and perform stochastic gradient descent updates. To achieve quasi-Newton peformances we adopted the Adagrad [3] algorithm to obtain an approximation of the diagonal of the

Figure 1: The Mechanical Turk interface.

Figure 2: The Mechanical Turk interface with a turker's predictions.

Figure 3: The overall architecture of our system.

Hessian as

$$\mathbf{H} = \sigma\mathbf{I} + \sum_{n=1}^{t-1} \operatorname{diag}(\mathbf{g}_n\mathbf{g}_n^\top), \tag{4}$$

where $\sigma$ is a small initialization term for numerical stability, and perform parameter upgrade as

$$\boldsymbol{\theta}_t = \boldsymbol{\theta}_{t-1} - \rho\mathbf{H}^+\mathbf{g}_t, \tag{5}$$

where $\rho$ is a predefined learning rate.

We took advantage of parallel computing by distributing the data over multiple machines and performing gradient computation in parallel, as it only involves summing up the per-datum gradient. As the data is too large to fit into the memory of even a medium-sized cluster, we only keep the minibatch in memory at each iteration, with a background process that pre-fetches the next minibatch from disk during the computation of the current minibatch. This enables us to perform efficient optimization with an arbitrarily large dataset. The overall architecture is visualized in Figure 3.

In terms of the libraries we used, we mainly adopted the Python + numpy framework for scientific computation (which is on par with commercial scientific computation software as long as it is based on an optimized BLAS library, which is most likely the case), which is fully open-source. The code runs on single machines, as well as over different machines in a distributed way. For distributed computation, we used the OpenMPI library and MPI4py as a Python interface, both of which are often used for scientific computation and only requires secure connections between machines to work.

We made the assumption that the machines will be up running during the whole computation time, and did not consider machine failures as in the Google system [2]. We believe that this design assumption is reasonable for single computers or medium-sized clusters, allowing us to simplify the implementation.

For the image features, we followed the pipeline in [6] to obtain over-complete features for the images. Specifically, we extracted dense local SIFT features, and used Local Coordinate Coding (LCC) to perform encoding with a dictionary of size 16K. The encoded features were then max pooled over 10 spatial bins: the whole image and the $3 \times 3$ regular grid. This yielded 160K feature dimensions per image, and a total of about 1.5TB for the training data in double precision format. The overall performance is 41.33% top-1 accuracy and a 61.91% top-5 accuracy on the validation data, and 41.28% and 61.69% respectively on the testing data. For the computation time, training with our toolbox took only about 24 hours with 10 commodity computers connected on a LAN. Our toolkit is implemented in Python and will be publicly available open-source at [hidden for double-blind review].

## 3   More on Confusion Matrix Estimation

Given a classifier, evaluating its behavior (including accuracy and confusion matrix) is often tackled with two approaches: using cross-validation or using a held-out validation dataset. In our case, we note that both methods have significant shortcomings. Cross-validation requires retraining the classifiers multiple rounds, which may lead to high re-training costs. A held-out validation dataset usually estimates the accuracy well, but not for the confusion matrix $\mathbf{C}$ due to insufficient number of validation images. For example, the ILSVRC challenge has only 50K validation images versus 1 million confusion matrix entries, leading to a large number of incorrect zero entries in the estimated confusion matrix.

| Smoothing | Source | Perplexity |
|---|---|---|
| | training | 94.69 |
| Laplace | validation | 80.52 |
| | unlearned | 46.95 |
| | training | 214.30 |
| Kneser-Ney | validation | 68.36 |
| | unlearned | **46.27** |

Table 1: The perplexity (lower values preferred) of the confusion matrix estimation methods on the testing data.

(a) Training      (b) Validation      (c) Unlearned      (d)

Figure 4: (a)-(c): Visualization of the zero estimations (averaged over $4 \times 4$ blocks for better readability) for non-zero testing entries obtained from multiple sources. (d): the proportion of zero estimations.

Instead of these methods, we approximate the classifier's leave-one-out (LOO) error on the training data with a simple gradient descent step to "unlearn" each image to estimate its LOO prediction, similar to the early unlearning ideas [4] proposed for neural networks. We will focus on the use of multinomial logistic regression, which minimizes $\mathcal{L}(\boldsymbol{\theta}) = \lambda\|\boldsymbol{\theta}\|_2^2 - \sum_{i=1}^{M} \mathbf{t}_i \log \mathbf{u}_i$, where $\mathbf{t}_i$ is a 0-1 indicator vector where only the $y_i$-th element is 1, and $\mathbf{u}_i$ is the softmax of the linear outputs $u_{ij} = \exp(\boldsymbol{\theta}_j^\top \mathbf{x}_i)/\sum_{j'=1}^{K} \exp(\boldsymbol{\theta}_{j'}^\top \mathbf{x}_i)$, with $\mathbf{x}_i$ being the feature for the $i$-th training image.

Specifically, given the trained classifier parameters $\boldsymbol{\theta}$, it is safe to assume that the gradient $\mathbf{g}(\boldsymbol{\theta}) = \mathbf{0}$. Thus, the gradient for the logistic regression loss when removing a training image $\mathbf{x}_i$ could be computed simply as $\mathbf{g}_{\backslash \mathbf{x}_i}(\boldsymbol{\theta}) = (\mathbf{u}_i - \mathbf{t}_i)\mathbf{x}_i^\top$. Also, notice that the accumulated matrix $\mathbf{H}$ obtained from Adagrad serves as a good approximation of the Hessian matrix[1], allowing us to perform one step quasi-Newton least-square update as

$$\boldsymbol{\theta}_{\backslash \mathbf{x}_i} = \boldsymbol{\theta} - \rho' \mathbf{H}^+ \mathbf{g}_{\backslash \mathbf{x}_i}. \tag{6}$$

Note that we put an additional step size $\rho'$ instead of $\rho' = 1$ as would be the case for exact least squares. We set $\rho'$ to the value that yields the same LOO approximation accuracy as the validation accuracy. We use the new parameter $\boldsymbol{\theta}_{\backslash \mathbf{x}_i}$ to perform prediction on $\mathbf{x}_i$ as if $\mathbf{x}_i$ has been left out during training, and accumulate the approximated LOO results to obtain the confusion matrix. We then applied Kneser-Ney [5] smoothing on the confusion matrix for a smoothed estimation.

Table 1 gives the perplexity values of the various sources to obtain the confusion matrix from: the training data (without unlearning), the validation data, and our approach (named as "unlearned"). Two different smoothing approaches are also adopted to test the performance: Laplace smoothing and Kneser-Ney smoothing, with the former smoothes the matrix by simply adding a constant term to each entry, and the latter taking a more sophisticated approach and utilizing the bigram information (see [5] for exact math). In general, our approach obtains the best perplexity over all choices.

Figure 4 visualizes the confusion matrix entries that are non-zero for the testing data, but incorrectly predicted as zero by the methods. Specifically, the dark regions in the figure shows incorrect zero estimates, so the darker the matrix is, the worse the estimation is. We also compute the proportion of

zero estimates, defined as the number of non-zero testing entries that are estimated as zero, divided by the total number of non-zero testing entries. The matrix is averaged over $4 \times 4$ blocks for better visualization. Overall, matrices estimated from the training and validation data both yield a large proportion ($>$70%) of incorrect zero entries due to over-fitting and lack of validation images respectively, while our method gives a much better estimation with incorrect zero entries $<$25%. Note that the problem of the remaining sparsity is further alleviated by the smoothing algorithms.

## Footnotes

[1]See supplementary material for details. In practice, we tested the Adagrad $\mathbf{H}$ matrix and the exact Hessian computed at $\boldsymbol{\Theta}$, and found the former to actually perform better, possibly due to its overall robustness.