[Reviews · NeurIPS 2013]

Submitted by Assigned_Reviewer_6

The authors present in this paper an algorithm for visual concept learning - given a few images of a concept (different concepts can be from different levels of abstraction) the algorithm learns to recognize other images from this concept. The algorithm combines object classification tools from machine vision and Bayesian generalization from cognitive science. The authors compare the generalization performances to other baseline methods and show there is a significant improvement when combining visual classifiers with the Bayesian cognitive model (as their algorithm does).
The authors also provide large-scale dataset for visual concept learning which is based on ImageNet images and human annotators.

I find the problem of visual concept learning interesting and relevant to the NIPS community.
The paper's quality is high: it is well written and very interesting to read.

There are few points that I think the authors should address:
- In section 3.2 the authors claim that their criteria (eq. 1) for choosing levels of nested concepts for generating their dataset result in sub-category, basic. super-basic and super-category levels in the taxonomy - I would like to see this claim validated.
I also think that the authors should cite Rosch et al. 1976 when they talk about basic-level categories, it would be interesting to connect between ideas from Rosch work to this work.

- Could there be some mix up in the equations of section 4.1?
Should equation 3 be P(h|X) or as it written P(X|h)?
I didn't understand why in equation 4 you have |h|^-N rather than |h|^-1 (maybe it should be P(X|h) and not P(x_i|h)) and also, it seems p(x_new|h) from line 226 doesn't fit equation 4 which is a bit confusing (I guess it should be p(x_new\inC|h).

- the description of the extension of [5] (HB) is unclear in my opinion. Shouldn't the subtree maintain the accuracy over the 5 example images rather than the query images (line 345)?
Summary: Well written paper about and interesting subject (concept learning) that is addressed using both tools from machine vision and cognitive science. I think it should be accepted.

Submitted by Assigned_Reviewer_7

This paper studies an interesting problem of visual concept learning, particularly paying attention to determine the degree to which a concept should be generalized. For this purpose, the work first focuses on how to generate a large-scale dataset of concepts at various levels of abstraction from ILSVRC and collect human judgements on AMT, and then presents Bayesian concept learning algorithm to handle perceptual uncertainty. The problem of learning concepts of various levels of abstraction is interesting. However, this reviewer finds that the algorithm presentation is unclear and lacks intuitive explanations. In addition, the experimental evaluation is somewhat weak because it doesn’t compare with strong baselines.

More detailed comments are listed as follows:

1. In Eq. (4) and (5), is I(.) an indicator function? Please clarity.

2. In Eq. (4), I(x_i \subseteq h) implies that $h$ is a set and $x_i$ is also a set. This is confusing to this reviewer. Why is $x_i$ a set? In contrast, in Eq. (5), I(\hat{y}_i \in h) shows that $\hat{y}_i$ is an item. Please clarify.

3. In Eq. (5), it seems that the summation is only effective for the first term, i.e. A_{j\hat{y}_i}. The remaining part is irrelevant to index $j$. Therefore the summation of A_{j\hat{y}_i} becomes one column vector. Is this what you want to get? I cannot understand the motivation of this equation.

4. It seems there is a typo in the last line of page 5. “… the true leaf node is $j$ given the classifier output being $j$”. Please check if you do want to put two $j$.

5. In line 243, page 5, the sentence “for example, as the number of examples that are all Dalmatians increases, it becomes increasingly likely that the concept is just Dalmatians and not dogs in general even though both are logically possible, …” is true. But what does this mean to the hypothesis size? The hypothesis is quite confusing in the paper. It seems that the authors define the hypothesis size as the number of samples belonging to a hypothesis. How can this be defined in real world, given that the number of potential images belonging to any concept can be infinitely large? Is the number of samples in ImageNet a reasonable way to define the hypothesis size?

6. Given so many unclear places in the algorithm presentation, it is quite hard to understand the proposed algorithm and know why it can really work, although the algorithm presentation is just of high level.

7. The baseline methods are not sufficiently strong and convincing. For example:
a) For the naïve vision approach, only GIST feature is used, whereas the proposed algorithm uses a overcomplete feature set of 160K dimensions. This is unfair to compare the two algorithms.
b) The paper mentions the latest work of deep neural network [10, 9] in Section 4.3. Why not choose DNN or its variant as a baseline method? Simply because it requires more training data? If DNN could output the proposed algorithm given that the training data can be easily collected from ImageNet, I would doubt the value of this work.
Summary: The paper studies an interesting problem of learning concepts at various levels of abstraction and the authors did put great efforts in constructing a large-scale concept learning dataset with human judgements collected on AMT. However, the algorithm presentation is unclear and lacks intuitive explanations. Also the experimental evaluation is unconvincing as some baseline method is implemented in a too simple way and some stronger baselines are not compared.

Submitted by Assigned_Reviewer_9

This paper presents a method to embed classification confidence into concept learning, and provides a dataset to evaluate it.

1. The proposed dataset is well constructed, and will make concrete contributions to category classification.

2. The idea of adding classification confidence into the system makes perfect sense, and provides a reasonable and practical way to approximate non-perceptual word learning method.

3. The paper devises a effective way to compute confusion matrix with limited data, facilitating the proposed algorithm.

4. The evaluation is well conducted, where the proposed algorithm is compared to nearest neighbor approaches and ideal non-perceptual word learning, showing a good performance.

-----

1. There might be a few typos in the paper:
In equation 4, the right hand side should be |h|^(-1) instead of |h|^(-N).
In equation 5, should the last term be I(j \in h) instead of I(y_i \in h)? Because otherwise the 1/|h|*I term can be move outside summation, and then the summation of confusion matrix does not make sense.
In line 269, Aj,i should be the confusion of i and j, not j and j.
These typos can be quite misleading to readers, so the paper should be check again for the final version.

2. The way classification confidence is added to the system is through confusion matrix in this paper. What if just use the confidence itself? For example, can the equation 5 be change to \sum_j confidence(j | x_i) 1/|h| I(j \in h)? How would this formulation work compared to the proposed method?
Summary: Overall, this paper proposes a effective way of concept learning, and a dataset to test on. It would be great if the typos are fixed and justification are given on why the proposed method is the best way to utilize classification confidence.
Author Feedback

Author rebuttal: We sincerely thank the reviewers for their time and valuable comments. The main strength of our paper is to propose a model that performs visual recognition using principles that model human concept learning, addressing a problem that is different from current image classification tasks but closer to actual human learning behavior. We also provided the first dataset for large scale experimentation of computational visually grounded concept learning, which future research may benefit from. Below we would like to offer our rebuttals and clarifications to certain points in the reviews: in particular we dispute the assertion in the reviews that there are missing baselines, or that deep learning may inherently solve our problem off-the-shelf.

== AR 6 ==

AR6 inquires about the way we choose levels in the ImageNet taxonomy. Although we experimented with many different methods for generating a concept hierarchy from ImageNet, we settled on the current one because upon examination, it created the most coherent set of hierarchically nested concepts (see Figure 2 for an example). Also, a previous version of the experiment using a different method for generating a concept hierarchy yielded very similar results, which suggests that the results may be robust to the concept generation method. Validation may be carried out by asking people to look at the levels and confirming their correctness, which is similar to what we did.

There is some slight confusion for the equations in section 4.1 (we would like to apologize for the typos, which would certainly be fixed). Equation 3 is the probability of generating an example from a hypothesis, thus P(\mathcal{X} | h). P(h | \mathcal{X}) is then proportional to Equation 3 times the prior P(h) [see lines 229-231]. Equation 4 samples one single example (|h|^-N should be |h|^-1), considering the size principle.

The probability of a new instance being a member of a hypothesis (defined in [lines 226-228]) differs in that it does not need the size principle. Intuitively, using a Dalmatian as an example when talking about animals is less probable than when talking about dogs (i.e. the size principle in generating examples), but a Dalmatian is definitely a member of both the animal and dog concepts. We think using the same P() notation for examples and new instances (equations 3 and 5) may have made the probabilities a little confusing, and will change the notations for better clarification.

== AR7 ==

We respectfully disagree with AR7 regarding the baselines, esp. the DNN approach.

(A) As stated in the first paragraph, we are proposing and solving a fundamentally different problem from conventional classification. In a sense, our task is to identify the correct level from all of the possible categories that a set of images may refer to, whereas conventional classification is to identify the most specific category for a single image. As DNN only solves the latter problem, it is not directly comparable to our method, and thus it is a tangential issue to the main thesis of the paper. Thus, we respectfully disagree that not using DNN weakens the value of the paper. At a high level, classification methods (such as DNN) provide the perceptual information in our concept learning framework. Our framework is compatible with any perceptual classifier, and thus, the current best perceptual classifier can always be plugged in to improve the performance of a method within our framework.

We do respect and are excited by the deep learning techniques [lines 293-296]. On a separate line of work, we are in the process of investigating different image classifiers including DNN, but improving the leaf node accuracy (using e.g. DNN) is again orthogonal to this paper, and is not the main interest here. We would be happy to add a few sentences to the paper for better clarification.

(b) The reason we used GIST for the naive vision approach [lines 320-322] is that, in fact, nearest neighbor with 160k-dim features works slightly worse empirically than GIST, because the effectiveness of simple Euclidean distances diminishes due to curse of dimensionality.

AR7 also inquires about the hypothesis size. The hypothesis sizes are computed as the number of leaf node classes under the corresponding hypothesis [lines 260-264]. We agree with the reviewer that different classes may have different frequencies in the real world, but counting the exact frequencies would be a dataset collection issue. ImageNet provides a “good enough” approximation of the frequency of object classes in the world. However, we appreciate the reviewer’s concern and hope better data sets are released in the future. Regardless, this concern does not affect the main thesis of the paper, and any better data sets could be used to improve the results of our framework.

Please kindly check the AR6 section for clarification on section 4.1.

== AR 9 ==

AR9 inquires about using confidence instead of confusion matrix. We actually tested and the latter works much better. The reason is that the classifier produces over-confident values due to overfitting the training data, and the confidence scores do not accurately match the actual classification probability. The confusion matrix estimated in the proposed way [lines 297-307] provides a more unbiased estimation.

Please kindly check the AR6 section for clarification on section 4.1.

== Misc ==

(AR6) we thank the reviewer for pointing out Rosch et al. 1976 and would add it to the discussions and citations.

(AR6) extension of [5] (HB): yes, the subtree maintains the accuracy over the example images. Sorry for the typo.

(AR7) eqn 4: I(.) is an indicator function. The \subseteq should actually be \in.

(AR7, AR9) bottom line [269] of page 5: it should be changed to "...the true leaf node is j given the classifier output being i".

Finally, we thank the reviewers again for the time and consideration.